# Clinical epidemiological characteristics and antibiotic sensitivity of *Escherichia coli* urinary tract infection

Zhi Wen*, Jiwei Jin, Yonghong Chen, Chengjuan Zhao, Chengchun Xu, Jun Liu, Binglei Ge*

Department of Laboratory Medicine, The Affiliated Xuancheng Hospital of Wannan Medical College, Xuancheng, Anhui, China

* ngahlgdx@163.com (ZW); bigwolf777@163.com (BG)

## Abstract

### Introduction

Urinary tract infections are common types of infections around the world, and most urinary tract infections are caused by *Escherichia coli* (*E. coli*). In order to better understand the clinical characteristics of *E. coli* in urinary tract infections and to guide empirical treatment, we conducted this study.

### Methods

*E. coli* in hospitalized patients with urinary tract infection in 2023 was retrospectively analyzed. Urine culture was determined and analyzed for patients with urinary tract infections admitted to Xuancheng People's Hospital from 01/01/2023–31/12/2023. More than $10^5$ cfu/ml in the urine culture supernatant is of great significance to urinary tract infection. According to needs, the identification and drug sensitivity tests were carried out using standard laboratory technology and automated system of France, VITEK 2 Compact, and the Clinical Laboratory Standards Institute (CLSI) standard was adopted.

### Results

A total of 401 strains were isolated from patients with urinary tract infections, including 62 hospital-acquired infections and 339 community acquired infections. Hospital-acquired infections were mainly ESBL-positive bacteria, and the general hospitalization time was about one month. Community acquired infections were mainly ESBL-negative bacteria. The hospital stay was about 10 days. The antibiotic susceptibility profiles were as follows: tigecycline (99.8%), imipenem (99.3%), ertapenem (99.3%), amikacin (98.3%), piperacillin/tazobactam (92.3%), cefxitin (87.3%), cefoperazone/sulbactam (85.8%), amoxicillin/clavulanate (76.6%), cefepime (76.3%),

**Data availability statement:** All relevant data are within the paper and its Supporting information files.

**Funding:** The study was funded by the Anhui Province Health Commission Scientific Research Project(Award Number:AHWJ2022c054).

**Competing interests:** The authors have declared that no competing interests exist.

ceftazidime (75.6%), ceftriaxone (65.8%), trimethoprim/sulfamethoxazole (64.1%), cefuroxime sodium (56.6%), cefuroximethoxazole (56.6%), and levofloxacin (21.9%).

## Conclusion

Urinary tract infections caused by *E. coli* are predominantly community acquired, accounting for 84.5%(339/401), and most cases involved ESBL-negative strains. Therefore, third-generation cephalosporins remain the preferred choice for empirical treatment, while quinolones and second-generation cephalosporins are not recommended. Subsequently, the treatment regimen can be de-escalated based on confirmed infection type and patient condition, an approach that has been associated with reduced hospitalization.

## Introduction

*Escherichia coli* (*E. coli*) is a common clinical Gram-negative bacillus and is also a normal intestinal bacterium. This pathogen can cause bloodstream infections and urinary tract infections in certain circumstances, such as window infections caused by diarrhea or dysentery, immunosuppression or catheterization [1]. *E. coli* have been common in hospital-acquired and community acquired infections over the past few decades and are almost the main pathogen [2–4].

Urinary tract infections (UTIs) occur when pathogens can enter the urinary tract system and exceed $10^5$ cfu/mL in the urine. UTIs was considered the second common cause of infectious diseases, following respiratory tract infections, and account for more than seven million office visits and more than one million visits to emergency departments in the United States, necessitating 100,000 hospitalizations [5,6]. According to previous studies [7,8], urinary tract infection accounts for about 40% of hospital-acquired infections and bacteria accounts for about 50%, which can prolong hospital stays and increase patient morbidity and mortality [9,10]. Urinary tract infections can also be divided into two types in clinical practice, complex and non-complex urinary tract infections. UTIs that occur 3 or more times a year and UTIs that occur 2 or more times in less than 6 months are considered complex UTIs [11,12], which are the main challenges in the treatment of UTI patients. Most urinary tract infections are caused by bacteria. The most common gram-negative bacteria are Enterobacteriaceae, mainly *E. coli*, and Staphylococcus are mainly among Gram-positive bacteria. *Uropathogenic E. coli* (UPEC) is a group of extra intestinal pathogenic *E. coli*. It is a common pathogenic *E. coli* originating from the intestine and can be transmitted through the oral-fecal pathway, contaminated food, etc [13]. The clinical data of 401 cases of urinary culture *E. coli* who were treated with duplicate strains at Xuancheng City People's Hospital from January 2023 to December 2023 (repeated strains have been removed) were retrospectively analyzed to inform optimal antibacterial treatment strategies and public health interventions.

## Methods

### Ethics

This study included *E. coli* isolates cultured from urine samples of hospitalized patients form 01/01/2023–31/12/2023. Ethical approval was obtained from the Ethics Committee of Xuancheng City People's Hospital (approval number: 2022-pjky005–01). Written informed consent was not required as the study used bacterial isolates collected as part of routine clinical care, and all patient data were anonymized and de-identified prior to analysis.

### Sample culture and antibiotic sensitivity testing of isolates

When the specimens were collected, the patient was hospitalized, the specimens sent for examination were urine and *E. coli* were cultured. Specimens that met all the above conditions were included in this study. This study found *E. coli* which caused urinary tract infection, by culturing urine in hospitalized patients from 01/01/2023–31/12/2023. All collected *E. coli* were tested for drug sensitivity (using VITEK 2; Merrier). Collected basic information (including gender, age, etc.) of the patient based on the information of the specimen in which *E. coli* has been cultured. According to whether *E. coli* has an Extended-Spectrum β-Lactamases (ESBL), it is divided into ESBL-positive and ESBL-negative groups. The differences in age, gender, and hospitalization time of hospitalization between the two groups were analyzed. At the same time, based on the patient's hospitalization data, the infection types were divided into two groups: hospital-acquired infection and community acquired infection. The clinical data of all cultured *E. coli* patients (including gender, age, hospitalization time, etc.) were collected and analyzed in the whole year of 2023. The drug sensitivity characteristics of *E. coli* in different infection types were explored.

Under the direction of nursing personnel, the patient collected urine samples in sterile containers and promptly submitted them to the laboratory for analysis within a 2-hour timeframe. Subsequently, 10 μL of the urine specimens were aseptically inoculated onto blood agar plates and then subjected to an incubation period at 35–37°C in an atmosphere containing 5% carbon dioxide for 48–72 hours. Following incubation, observation of colony morphology, Gram staining, and colony quantification were performed. Bacterial species identification and antimicrobial susceptibility testing were conducted using the VITEK 2; bioMérieux Compact system. All operation processes comply with operation norms and procedures.

The quality control strains used for VITEK 2 were *E. coli* ATCC 25922, *P. aeruginosa* ATCC 27853, and *S. aureus* ATCC 29213.

### Statistical analysis

We expressed continuous variables as mean ± SD, and the results of categorical data as numbers with percentages. We compared continuous variables using the Mann-Whitney U test and the Wilcoxon W test, and compared categorical variables using the chi-square test. The results were considered statistically significant when $p < 0.05$. All statistical analyses were conducted using SPSS 23.0 software.

## Results

### General situation of research subjects

A total of 401 subjects to the patients that met the inclusion criteria were included in this study, including 135 cases in the ESBL-producing group and 266 cases in the ESBL-negative group. The mean age of the ESBL-positive group was 63.74 ± 17.54 years, and the mean age of the ESBL-negative group was 61.54 ± 18.43 years. After statistical test, there was no significant difference in age between the two groups (p = 0.252). The hospital stay in the two groups was further analyzed. The mean hospital stay in the ESBL-positive group was 16.98 ± 19.35 days, and the mean hospital stay in the ESBL-negative group was 10.48 ± 10.22 days. It was found that there was a significant difference in hospital stays

between the two groups, which is statistically significant (p < 0.001). Checkout time which is the number of days between the start of the patient's hospitalization and the laboratory culture of *E. coli* in the patient's urine, Statistical significance is also found between the two groups. Table 1 for details.

## Hospital-acquired infection and community acquired infection

Hospital-acquired infections are mostly exogenous infections. Indirect transmission is the most common mechanism of hospital-acquired infection pathogens. They are mechanically transferred through the hands and clothes of health workers, whose hands or clothes are infected by themselves or other patients [14]. Community acquired infection, also known as community infection, refers to infections obtained by an individual in a community environment, rather than in a specific medical institution (such as hospitals, clinics, long-term care institutions, etc.). Pathogens are obtained by exposure to daily activities of the community (such as work, study, etc.). The occurrence of infection has no clear connection with diagnosis and treatment activities, hospitalization, visits, etc. in medical institutions. An infection that is already present at the time of admission or an infection that is obtained in the community but is in the latent period after admission, and subsequently onset [15].

A total of 401 patient samples were included in this study. According to the location of the infection, it can be divided into 62 hospital-acquired infection group and 339 community acquired infection group. Hospital-acquired infections in the urinary tract, *E. coli* mainly produces Extended-Spectrum β-lactamase (ESBL)-positive strains, accounting for 67.74% (42/62). In contrast, ESBL-negative strains were mainly among community acquired infections, accounting for 72.56% (234/339). Statistical analysis showed that the difference in ESBL positive rates between the two groups was significant (p < 0.001). See Table 2 for detailed data.

## Antimicrobial susceptibility of *E. coli* isolates

The *E. coli* strain isolated from urinary tract infections was subjected to drug sensitivity testing using the VITEK 2 Compact analyzer manufactured by the French company bioMérieux. The results of the antibiotic sensitivity analysis were

**Table 1. Distribution of ESBL positive group and ESBL negative group.**

| | ESBL"+" (n = 135) | ESBL"-" (n = 266) | Statistics | p value |
|---|---|---|---|---|
| Male/Female (n) | 45/90 | 53/213 | $X^2 = 8.719$ | 0.005 |
| Age (years) | 63.74 ± 17.54 | 61.54 ± 18.43 | F = 1.317 | 0.252 |
| Hospitalization time [a] (days) | 16.98 ± 19.35 | 10.48 ± 10.22 | F = 19.354 | <0.001 |
| Checkout time (days) | 8.29 ± 12.25 | 4.68 ± 10.42 | F = 9.535 | 0.002 |

[a]: Hospitalization time refers to the total duration (in days) from hospital admission to discharge for a single clinical episode, excluding time in emergency departments or outpatient clinics prior to formal admission.

**Table 2. Clinical characteristics of 401 patients.**

| | | Hospital-acquired infection (n = 62) | Community acquired infection (n = 339) | Statistics | p value |
|---|---|---|---|---|---|
| ESBL | + (n,%) | 42 (67.7) | 93 (27.4) | $X^2 = 38.134$ | <0.001 |
| | - (n,%) | 20 (32.3) | 246 (72.6) | | |
| Gender | Male(n,%) | 17 (27.4) | 81 (23.9) | $X^2 = 0.353$ | 0.525 |
| | Female(n,%) | 45 (72.6) | 258 (76.1) | | |
| Age (years) | | 64.53 ± 17.35 | 61.87 ± 18.28 | F = 1.129 | 0.289 |
| Hospitalization time (days) | | 29.84 ± 23.84 | 9.53 ± 8.66 | F = 143.771 | <0.001 |

presented in Table 3. The *E. coli* isolates exhibited notable susceptibility (>80%) to a wide range of antibiotic classes, such as aminoglycosides, cephalosporins, and carbapenems. Particularly, imipenem (IPM) and ertapenem (ETP) displayed remarkable efficacy. Cephalosporin antibiotics demonstrated notable sensitivity levels, with second-generation cephalosporins such as cefuroxime sodium (CXM) and cefuroxime (CXMA) exceeding 50% sensitivity, third-generation cephalosporins like ceftriaxone (CRO) and ceftazidime (CAZ) showing sensitivities between 60% and 80%, and fourth-generation cephalosporins such as cefepime (FEP) approaching 80% sensitivity. In contrast, Quinolones such as levofloxacin (LEV) exhibited the highest resistance rate to our isolates rate at 78.1% warranting heightened attention from clinicians. Carbapenem-Resistant Enterobacterales (CRE): Resistance to nearly all β-lactams, including carbapenems. Often mediated by carbapenemases (KPC, NDM, OXA-48, VIM, IMP). In this study, several CRE strains appeared, and the presence of these strains also indicates that this multi-drug-resistant pattern of *E. coli* is also distributed in urine, even if the probability is very low, it should not be ignored.

This study revealed the sensitivity of *E. coli* to antibiotics in urinary tract infection by classifying the infection status of patients. The results of the study showed that there was no significant statistical difference in the sensitivity of six antibiotics, ETP, cefoxitin (FOX), IPM, trimethoprim/sulfamethoxazole (SXT), piperacillin/tazobactam (TZP), and amoxicillin/clavulanic acid (AMC) in hospital-acquired infection and community acquired infection (p > 0.05). However, antibiotics such as cephalosporin antibiotics, AMK, LEV, CLS, TGC, etc. Statistics showed that there were differences among different infection groups, and the differences were statistically significant. (p < 0.05). See Table 4 for detailed data.

## Discussion

This study found that *E. coli* cultured in urine was categorized into ESBL-positive and ESBL-negative groups based on the production of Extended-Spectrum β-lactamase. There were statistical differences between the groups in hospitalization time and time to detection (p < 0.001). The ESBL-positive group had longer hospitalization and detection times, which was similar to the conclusions drawn by Marc J M Bonten in 2018 [16]; However, there was no significant differences between the groups in terms of gender composition ratio and age factors (p = 0.17, p = 0.353).

Upon combining the patient's basic information with their infection status, it was observed that *E. coli* infections in urine were predominantly community acquired, accounting for 84.5% (339/401), The majority of the community acquired infections involved ESBL-negative bacteria. In contrast, *E. coli* infected in hospitals were primarily caused by ESBL-positive bacteria, accounting for 67.74% (42/62). A statistically significant difference was noted between the two groups (p < 0.001); Additionally, the average hospital stay for patients with hospital-acquired infection group was close to one month, significantly higher

**Table 3. Susceptibility of 401 *E. coli* isolates to antimicrobial agents.**

| antibiotic | Sensitivity (%) | antibiotic | Sensitivity (%) |
|---|---|---|---|
| AMK | 98.3 | CXMA | 56.6 |
| CRO | 65.8 | CLS | 95.8 |
| ETP | 99.3 | SXT | 64.1 |
| CAZ | 76.3 | FEP | 76.3 |
| FOX | 87.0 | TGC | 99.8 |
| IPM | 99.3 | TZP | 92.3 |
| LEV | 21.9 | AMC | 76.6 |
| CXM | 56.6 | | |

Legend: AMK: amikacin; CRO: ceftriaxone; ETP: ertapenem; CAZ: ceftazidime; FOX: cefoxitin; IPM: imipenem; LEV: levofloxacin; CXM: cefuroxime sodium; CXMA: cefuroxime axetil; CLS: cefoperidone/sulbactam; SXT: trimethoprim/sulfamethoxazole; FEP:cefepime; TGC: tigecycline; TZP: piperacillin/tazobactam; AMC: amoxicillin/clavulanic acid.

**Table 4. Antimicrobial susceptibility and differences in each antibiotic under different infection type.**

| Antibiotic | S OR R | Infection type | | Mann-Whitney U | Wilcoxon W | Z | p value |
|---|---|---|---|---|---|---|---|
| | | Hospital-acquired (n = 62) | Community acquired (n = 339) | | | | |
| AMK | S (n,%) | 59 (95.2) | 335 (98.8) | 10121.5 | 67751.5 | −2.036 | 0.042 |
| | R (n,%) | 3 (4.8) | 4 (1.2) | | | | |
| CRO | S (n,%) | 20 (32.3) | 244 (72.0) | 6314.0 | 63944.0 | −6.082 | <0.001 |
| | R (n,%) | 42 (67.7) | 95 (28.0) | | | | |
| ETP | S (n,%) | 61 (98.4) | 337 (99.4) | 10401.5 | 68031.5 | −0.858 | 0.391 |
| | R (n,%) | 1 (1.6) | 2 (0.6) | | | | |
| FEP | S (n,%) | 30 (48.4) | 273 (80.5) | 6896.0 | 64525.0 | −5.737 | <0.001 |
| | R (n,%) | 32 (51.6) | 66 (19.5) | | | | |
| FOX | S (n,%) | 54 (87.1) | 295 (87.0) | 10507.0 | 68137.0 | −0.004 | 0.997 |
| | R (n,%) | 8 (12.9) | 44 (13.0) | | | | |
| IPM | S (n,%) | 61 (98.4) | 337 (99.4) | 10401.0 | 68031.5 | −0.858 | 0.391 |
| | R (n,%) | 1 (1.6) | 2 (0.6) | | | | |
| LEV | S (n,%) | 10 (16.1) | 78 (23.0) | 8753.0 | 66383.0 | −2.263 | 0.024 |
| | R (n,%) | 52 (83.9) | 261 (77.0) | | | | |
| CXM | S (n,%) | 18 (29.0) | 210 (61.9) | 6564.5 | 64194.5 | −5.369 | <0.001 |
| | R (n,%) | 44 (71.0) | 129 (38.1) | | | | |
| CXMA | S (n,%) | 18 (29.0) | 210 (61.9) | 6564.5 | 64194.5 | −5.369 | <0.001 |
| | R (n,%) | 44 (71.0) | 129 (38.1) | | | | |
| CLS | S (n,%) | 56 (90.3) | 328 (96.8) | 9834.5 | 67464.5 | −2.303 | 0.021 |
| | R (n,%) | 6 (9.7) | 11 (3.2) | | | | |
| SXT | S (n,%) | 36 (58.1) | 221 (65.2) | 9747.0 | 67377.0 | −1.092 | 0.275 |
| | R (n,%) | 26 (41.9) | 118 (34.8) | | | | |
| CAZ | S (n,%) | 25 (40.3) | 281 (82.9) | 6045.5 | 63675.5 | −7.179 | <0.001 |
| | R (n,%) | 37 (59.7) | 58 (17.1) | | | | |
| TGC | S (n,%) | 61 (98.4) | 339 (100) | 10339.5 | 67969.5 | −2.338 | 0.019 |
| | R (n,%) | 1 (1.6) | 0 (0) | | | | |
| TZP | S (n,%) | 55 (88.7) | 315 (92.9) | 10057.5 | 67687.5 | −1.163 | 0.245 |
| | R (n,%) | 7 (11.3) | 24 (7.1) | | | | |
| AMC | S (n,%) | 43 (69.4) | 264 (77.9) | 9494.5 | 67124.5 | −1.635 | 0.102 |
| | R (n,%) | 19 (30.6) | 75 (22.1) | | | | |

than that of patients with community acquired infections (29.8 days vs. 9.5 days). This suggests that hospital-acquired infections will lead to longer anti-infection treatment, which poses a huge challenge to clinical management.

Statistics on drug sensitivity of *E. coli* indicate that the sensitivity rate of strains causing urinary tract infections to most antibiotics can reach 80%. Specifically, carbapenem antibiotics (such as ertapenem, imipenem) and tetracycline antibiotics (like tigecycline) exhibit sensitivity rates exceeding 95%, while cephalosporin antibiotics show a sensitivity rate above 50%. Quinolones such as levofloxacin (LEV) exhibited the highest resistance rate to our isolates rate at 78.1% warranting heightened attention from clinicians. Stratifying by infection type revealed significant differences in sensitivity among cephalosporin, amikacin, levofloxacin, cefoperazone/sulbactam, piperacillin/tazobactam, and tigecycline, whereas no statistical variance was noted among the remaining antibiotics. Cephalosporin antibiotics are highly effective for treating community acquired infections and remain a preferred option based on clinical practice [17]. For hospital-acquired patients with *E. coli* infections, antibiotics with higher sensitivity rates, including aminoglycoside amikacin, carbapenem antibiotics (ertapenem, imipenem), cefoperidone/sulbactam, and tigecycline are recommended.

In this study, the resistance rate of trimethoprim/sulfamethoxazole due to *E. coli* approached 40%, aligning with prior literature from South Korea and China [18,19]. Conversely, in European countries such as Germany, France, and the United Kingdom, the resistance rate of *E. coli* to trimethoprim/sulfamethoxazole in urinary tract infections typically ranged between 10% and 20%. The lowest rate observed was 4.9% in Finland, whereas the highest was 26.7% in Portugal [20,21]. ESBL testing remains a crucial component of *E. coli* drug sensitivity experiments. It has been consistently reported that the global ESBL-positive rate among *E. coli* exhibits an upward trajectory. Prior to 2010, the ESBL-positive rate in most countries stood at 5%–10% [22–25], while for *E. coli* associated with urinary tract infections, it surpassed 20% [26]. In the current sample collection, the ESBL-positive rate of *E. coli* reached 33.7% (135/401). This indicates a marked increase in the ESBL-positive rate over the past decade. Consequently, the rise in ESBL production among *E. coli* may also be intricately linked to its drug resistance mechanism. ESBL-producing among *E. coli* can integrate genes enhancing bacterial survival into chromosomal DNA through drug-resistant plasmids such as IncFII, IncN and IncII, thereby improving drug resistance in subsequent generations [27,28].

The formation of biofilms remains a significant risk factor for complex urinary tract infections caused by *E. coli* [29]. Various antibiotics exhibit distinct impacts on the formation of biofilms of *E. coli* within urinary tract infections. Research indicates that 48 hours post-bioflim development, cephalosporins, aminoglycosides and quinolones notably diminish biofilm biomass [30,31]. Cephalosporin-resistant bacteria inhibit biofilm formation by harboring CTX-M-2 and CTX-N-9 ESBL genes. Aminoglycoside antibiotics such as AMK, impede biofilm synthesis by disrupting bacterial ribosomal protein synthesis. Quinolones penetrate host cell membranes, accumulate within cells, inhibit DNA replication, and reduce biofilm resilience. Additionally, this study revealed statistically significant differences in the efficacy of these antibiotics between hospital-acquired and community acquired infections.

*E. coli* is a prevalent bacterium in urinary tract infections. This study, which collected data over a single year and conducted subsequent analysis, was limited by a brief sampling period, thereby failing to fully and accurately reflect the clinical characteristics of *E. coli* in urinary tract infections within this region. Furthermore, this study was limited to data collection from our hospital and did not incorporate information from other medical institutions, which undermined the comprehensive accuracy of *E. coli*'s antimicrobial susceptibility profiles within the region.

## Conclusion

In summary, urinary tract infections caused by *E. coli* remain a significant concern that cannot be ignored. Our finding indicate that *E. coli*-induced urinary tract infections predominantly occur community acquired infection and typically have a shorter duration. Conversely, ESBL-positive cases are primarily associated with hospital-acquired infections and exhibit a prolonged disease course. During antimicrobial therapy, extended-spectrum antibiotics such as AMK, TGC, and TZP should be prioritized. For clinical frontline healthcare professionals, understanding the clinical epidemiology and antibiotic resistance patterns of *E. coli* in urinary tract infections is crucial for early identification and appropriate treatment, ultimately reducing hospital stays and the incidence of complications.

## Supporting information

**S1 File. Distribution of ESBL positive group and ESBL negative group.**
(XLSX)

**S2 File. Clinical characteristics of 401 patients.**
(XLSX)

**S3 File. Antimicrobial susceptibility of 401 E. coli isolates; Antimicrobial susceptibility and differences in each antibiotic under different infection type.**
(XLSX)

## Author contributions

**Data curation:** Zhi Wen, Jiwei Jin, Yonghong Chen, Chengjuan Zhao, Chengchun Xu, Jun Liu, Binglei Ge.

**Funding acquisition:** Zhi Wen.

**Writing – original draft:** Zhi Wen.

**Writing – review & editing:** Zhi Wen, Jiwei Jin, Binglei Ge.

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
