## [Decision Letter · Decision Letter 0]

16 Jul 2025

Dear Dr.  Wen,

Thank you for submitting your manuscript to PLOS ONE. After careful consideration, we feel that it has merit but does not fully meet PLOS ONE’s publication criteria as it currently stands. Therefore, we invite you to submit a revised version of the manuscript that addresses the points raised during the review process.

We look forward to receiving your revised manuscript.

Kind regards,

Mabel Kamweli Aworh, DVM, MPH, PhD. FCVSN

Academic Editor

PLOS ONE

Journal Requirements:

“Anhui Provincial Health Commission Scientific Research Project”

5. Please include captions for your Supporting Information files at the end of your manuscript, and update any in-text citations to match accordingly. Please see our Supporting Information guidelines for more information: http://journals.plos.org/plosone/s/supporting-information .

6. We note that there is identifying data in the Supporting Information file renamed_2709f.xlsx and renamed_424be.xlsx. Due to the inclusion of these potentially identifying data, we have removed this file from your file inventory. Prior to sharing human research participant data, authors should consult with an ethics committee to ensure data are shared in accordance with participant consent and all applicable local laws.

-Location data

Additional Editor Comments:

Thank you for your submission. In addition to addressing the reviewers’ comments, please attend to the following editorial revisions to strengthen the clarity and consistency of your manuscript:

**Manuscript Formatting** : Please ensure that the revised manuscript includes **line numbers and page numbers** throughout to facilitate efficient review.**Scientific Nomenclature** : The species name *Escherichia coli* should be **italicized** wherever it appears in the manuscript. Specifically, correct formatting is needed in:Line 1 and line 3 of the **Introduction**Line 1 of the **Methods**All subsequent occurrences throughout the manuscript**Discussion Section** : Please avoid restating results in the **Discussion** section. Instead, focus on interpreting your findings, offering potential explanations, and discussing how they compare or contrast with previously published studies. This will help contextualize your contributions more effectively.**Limitations** : A concise summary of the **key limitations** of your study should be added as the **final paragraph of the Discussion** section. This enhances transparency and provides a foundation for future research directions.**Conclusion** : Please incorporate a standalone **Conclusion** section. It should:Summarize the **key findings** of your studyPresent **recommendations or implications** based on your results in the final paragraph

Reviewers' comments:

Reviewer's Responses to Questions

**Comments to the Author**

1. Is the manuscript technically sound, and do the data support the conclusions?

Reviewer #1: Yes

Reviewer #2: Yes

Reviewer #3: Partly

2. Has the statistical analysis been performed appropriately and rigorously?

Reviewer #1: Yes

Reviewer #2: N/A

Reviewer #3: No

3. Have the authors made all data underlying the findings in their manuscript fully available?

Reviewer #1: Yes

Reviewer #2: Yes

Reviewer #3: Yes

4. Is the manuscript presented in an intelligible fashion and written in standard English?

Reviewer #1: Yes

Reviewer #2: No

Reviewer #3: No

Reviewer #1: INTRODUCTION

The introduction is logical and flows well. The prevalence/epidemiological context are clearly stated. Also, the clinical relevance is well established. However, the followings should be clarified.

1. Clarity. “Under certain circumstances”. Specify some of the circumstances e.g., immunosuppression or catheterization.

2. Quantitative specification. “UTI is considered the second common cause of infectious diseases” …. Reference and state specific epidemiological data or global incidence rankings to support this claim.

3. UTI classification. … “UTI is classified as a complex and non-complex” … Add a brief definition of what constitutes a complex UTI for clarification to the readers.

4. Terminology precision. “Parenteral pathogenic” is a bit confusing. Change this to “extra intestinal pathogenic” if referring to UPEC’s ability to cause infections outside the gut.

5. Closing statement. … “to reveal their drug resistance and clinical epidemiological characteristics”. This can be ended with a stronger and precision phase such as “with the goal of guiding effective antimicrobial treatment plan and public health intervention”.

6. … “admitted from January 2023 to December 2023” …. Specify if this is a multicenter study or single hospital dataset.

METHODS

Ethics

1. Break into clearer sentences

2. Ensure formal tone and compliance with international ethics reporting standards.

For example, “This study included Escherichia coli isolates cultured from urine samples of hospitalized patients between January 1, 2023, and December 31, 2023. Ethical approval was obtained from the Ethics Committee of Xuancheng City People's Hospital (Approval Number: 2022-pjky005-01). Written informed consent was not required as the study used bacterial isolates collected as part of routine clinical care, and all patient data were anonymized and de-identified prior to analysis”.

Experimental procedure

1. Improve on the grammar and flow.

2. Change typographical error “form” to “from” (patients form 01/01/2023 to 31/12/2023).

Bacterial culture

The opening sentence is unclear. Remove or rewrite if it must be there.

Streamline method steps for clarity.

STATISTICAL ANALYSIS

1. Correct grammar and punctuations

2. Ensure tense consistency (past tense).

3. Clarify statistical methods and match with international reporting standards.

4. Ensure consistency in units (e.g. μL)

5. Spacing and software naming

RESULTS

General situation of research subjects

1. Clarity and formality

1. Use precise and formal phrasing e.g. subjects to patients/cases.

Statistical test to statistical analysis.

2. Avoid redundancy e.g. “statistical results show….. which is statistically significant is repetitive.

2. Numerical reporting

1. Report p-values consistently with standard formatting (e.g. P = 0.252, do not write in italics).

2. Remove parentheses around mean ± SD values except required.

3. Terminology consistency

I would suggest using hospital-acquired and non-hospital acquired infections instead of “in-hospital” and “out-of-hospital” for standardization.

4. Grammar and style

1. Improve sentence flow and eliminate awkward phrasing.

2. Clarify confusing logic. E.g. P>0.05 was significant. This contradicts standard significance thresholds. In other words, this does not signify significance, yet the statement claims the opposite. Confirm the intended meaning and adjust accordingly.

Drug susceptibility

1. Manufacturer name error. “Merrier” should be corrected to bioMérieux.

2. Redundancy and flow. The phrase ‘Escherichia coli exhibited high sensitivity to. Most antibiotics with rates exceeding 80% and some surpassing 90%...” is a bit repetitive. Group antibiotics into classes for better readability.

3. Scientific tone

1. Change drug sensitivity to antimicrobial susceptibility for precision.

2. State clearly which findings require clinical caution e.g. low fluoroquinolone sensitivity.

3. Ensure that drug abbreviations (e.g., AMK, ETP) are defined.

4. Highlight any multidrug resistance patterns briefly.

DISCUSSION

1. Grammar and language use

1. Rewrite the multiple redundant constructions

2. Inconsistent verb tenses (e.g. it was found that… to is...)

3. Change this phrase … “bacteria detection time to e.g. time to detection” for clarity.

2. Scientific tone and structure

Change the informal expressions to professional tone e.g. face more difficult anti-infection….

3. Statistical reporting. Use consistent phrasing for P-values and avoid redundancy.

4. Content clarity

1. The connection between biofilm formation and antibiotic resistance could be more directly and clearly explained.

2. Clarify the contrast between your findings and previous literature where appropriate.

Did you encounter any limitation in this study? If yes, indicate them.

REFERENCES

Ensure all references (e.g., [13] – [28]) are accurately cited in the reference list.

Reviewer #2: 1. Re-frame the title. Avoid the repeated use of characteristics

2. 2nd paragraph, Line4…” 40% of hospital infections” - do you mean hospital acquired infections? Clarify

3. Same paragraph: The most common gram-negative bacteria is Enterobacteriaceae, mainly E. coli, and Staphylococcus is mainly among Gram-positive bacteria. Replace ‘is’ with are. Uropathogenic Escherichia coli (UPEC) is a group of parenteral pathogenic Escherichia coli should be italicized

4. The subtitle “Experimental procedure” is perhaps more appropriately expressed as laboratory testing

5. Still under the same subtitle…“using the VITEK 2 Compact system of the French company Merrier).” Shorten it like......VITEK 2; Merrier). Also apply to other parts of the manuscript where this occurs.

6. Extended-Spectrum β-Lactamases- first use insert (ESBL) after it

7. “Hospitalization time” define; explain

8. “in-hospital infection and out-of-hospital infection.” Define these terms.

9. “characteristics of the infected population (including gender, age, hospitalization time, etc.)” Time of the year) not given. This will give temporality to the epidemiological data given

10. Bacterial culture- could be changed to: Sample culture and antibiotic sensitivity testing of isolates

11. “Research on bacterial culture and drug sensitivity test methods” – Delete

12. Under subtitle Statistical analysis “This study uses SPSS23.0” uses should be in the past tense. This should be applied across the methods section. English improvement is generally needed.

13. Checkout time in Table 1 should be defined

14. Under the subtitle “In-hospital infection and out-of-hospital infection” the first statement is a repetition. Delete

15. Subtitle “A Drug susceptibility of 401 cases of Escherichia coli”: make it simpler. As in: Drug susceptibility of Escherichia coli isolates

16. May wish to reframe “In contrast, quinolones like levofloxacin (LEV) exhibited the lowest sensitivity rate at 21.9%, warranting heightened attention from clinicians.” As: “Quinolones such as levofloxacin (LEV) exhibited the highest resistance rate to our isolates rate at 78.1% warranting heightened attention from clinicians.” To emphasize the resistance

17. Table 2: Clinical characteristics of 401 patients under different infection conditions. Delete” under different infection conditions”. Compute and add %s in the table

18. Table 3: Sensitivity rates of various antibiotics of 401 cases of Escherichia coli. Again, shorten and make simpler. Provide legend below giving the full name of the abbreviations in the table

19. intra-hospital and out-of-hospital infections – uniformity in terminologies used in essential. Is this the same as “in-hospital infection and out-of-hospital infection” used elsewhere in the text. It’s confusing.

20. “showed significant differences between different infected groups, and the difference was statistically significant detailed data.”. English usage issues

21. In the fourth paragraph under discussion, the statement “However, compared with the situation in European countries, the drug resistance rate of Escherichia coli to compound trimethoprim/ sulfamethoxazole has decreased in several European countries (17, 18).” There seems to be no comparison. Perhaps another English challenge.

22. The limitations of the study were not listed or discussed.

23. The very last but long last statement requires English editing.

Reviewer #3: The authors of this paper have identified an opportunity to correlate retrospective hospital case data, samples in a hospital biobank with hospitalization records for urinary tract infection as a result of Escherichia coli. The authors intended approach to associate the antibiotic susceptibility profiles for the E.coli strains and their corresponding Extended-Spectrum β-lactamase status with hospitalization cases and perhaps source of infection is a unique perspective, which if properly articulated can allow hospital systems develop protocols to reviewing antibiotic use or discontinued use within hospital systems.

The authors have however not clearly articulated their thoughts in the current report. The paper will benefit from a major revision across all sections of the paper from abstract to discussion. In the current state, the paper is hard to follow, requiring major editorial input. Some quick but not exhaustive list of highlights below.

This paper will benefit majorly from a total revamp and editorial assistance for grammar and consistency in readability.

**Do you want your identity to be public for this peer review?** For information about this choice, including consent withdrawal, please see our Privacy Policy

Reviewer #1: No

Reviewer #2: No

Reviewer #3: No

---

## [Author Response · Author response to Decision Letter 1]

26 Aug 2025

Point-by-point response to Referees’ comments

Reviewer #1:

INTRODUCTION

The introduction is logical and flows well. The prevalence/epidemiological context are clearly stated. Also, the clinical relevance is well established. However, the followings should be clarified.

1.Clarity. “Under certain circumstances”. Specify some of the circumstances e.g., immunosuppression or catheterization.

Response: Thank you for highlighting the need for specificity. We have revised the phrase 'under certain circumstances' to explicitly state relevant clinical contexts. The text now reads: “such as window infections caused by diarrhea or dysentery, immunosuppression or catheterization”. (line 46-47)

2.Quantitative specification. “UTI is considered the second common cause of infectious diseases” …. Reference and state specific epidemiological data or global incidence rankings to support this claim.

Response: Thank you for requesting epidemiological clarification. We have revised the statement to reflect authoritative sources and added quantitative context: following respiratory tract infections, and account for more than seven million office visits and more than one million visits to emergency departments in the United States, necessitating 100,000 hospitalizations” with new references. (line 52-54)

3. UTI classification. … “UTI is classified as a complex and non-complex” … Add a brief definition of what constitutes a complex UTI for clarification to the readers.

Response: Thank you for this constructive suggestion. We have corrected the word "recurrent" to "complex". (line 60)

3.Terminology precision. “Parenteral pathogenic” is a bit confusing. Change this to “extra intestinal pathogenic” if referring to UPEC’s ability to cause infections outside the gut.

Response: Thank you for this important clarification. We agree that 'parenteral pathogenic' may cause confusion in this context. The text has been revised in the revised manuscript. (line 64)

5. Closing statement. … “to reveal their drug resistance and clinical epidemiological characteristics”. This can be ended with a stronger and precision phase such as “with the goal of guiding effective antimicrobial treatment plan and public health intervention”.

Response: Thank you for your suggestion. We all agreed that your advice was very professional,and we have revised it in the revised manuscript. (line 70)

6. … “admitted from January 2023 to December 2023” …. Specify if this is a multicenter study or single hospital dataset.

Response: Thank you for highlighting this important detail. We confirmed that this is a single-center study. The revised text now explicitly states: The clinical data of 401 cases of urinary culture E. coli who were treated with duplicate strains at Xuancheng City People's Hospital from January 2023 to December 2023 (repeated strains have been removed) (line 67-68)

METHODS

Ethics

1. Break into clearer sentences

Response: Thank you for your reminder, we have revised the relevant sentences.

2. Ensure formal tone and compliance with international ethics reporting standards.

For example, “This study included Escherichia coli isolates cultured from urine samples of hospitalized patients between January 1, 2023, and December 31, 2023. Ethical approval was obtained from the Ethics Committee of Xuancheng City People's Hospital (Approval Number: 2022-pjky005-01). Written informed consent was not required as the study used bacterial isolates collected as part of routine clinical care, and all patient data were anonymized and de-identified prior to analysis”.

Response: Thank you for emphasizing ethical reporting rigor. We have revised it in the revised manuscript. (line 73-78)

Experimental procedure

1. Improve on the grammar and flow.

Response: Thanks for your reminder. We have improved the grammar of the manuscript.

2. Change typographical error “form” to “from” (patients form 01/01/2023 to 31/12/2023).

Response: Thank you for noting this oversight. The error has been corrected. (line 83)

Bacterial culture

The opening sentence is unclear. Remove or rewrite if it must be there. Streamline method steps for clarity.

Response: Thank you for your reminder. The opening sentence has been removed in the revised manuscript.

4.STATISTICAL ANALYSIS

1. Correct grammar and punctuations

Response: We have improved the grammar and punctuation in the revised manuscript.

2. Ensure tense consistency (past tense).

Response: We have modified the tense of the sentence in the article to keep it consistent.

3. Clarify statistical methods and match with international reporting standards.

Response: Thanks for your professional advice. We have matched the statistical methods used in this study and international reporting standards. (line 107-111)

4. Ensure consistency in units (e.g. μL)

Response: We have made changes in the revised manuscript. (line 97)

5. Spacing and software naming

Response: We have adjusted the spacing to make sure the correct software name was used. (line111)

RESULTS

General situation of research subjects

1. Clarity and formality

1. Use precise and formal phrasing e.g. subjects to patients/cases.Statistical test to statistical analysis.

Response: Thank you for your professional advice. We have modified it in the revised manuscript based on your suggestions. (line 114)

2. Avoid redundancy e.g. “statistical results show….. which is statistically significant is repetitive.

Response: Thank you for highlighting this redundancy. We have made changes in the revised manuscript. (line 117-119)

2. Numerical reporting

1. Report p-values consistently with standard formatting (e.g. P = 0.252, do not write in italics).

Response: Thank you for emphasizing this important formatting detail. We confirm all p-values are now reported consistently throughout the revised manuscript. (line 119, 123, 183, etc.)

2. Remove parentheses around mean ± SD values except required.

Response: Thank you for your reminder. We have removed parentheses at this position according to your suggestion. (line 116, 117, 120, and 121)

3. Terminology consistency

I would suggest using hospital-acquired and non-hospital acquired infections instead of “in-hospital” and “out-of-hospital” for standardization.

Response: Thank you for your suggestion. We have replaced "hospital-acquired" with "in-hospital" and “non-hospital acquire” with “out-of-hospital” in the whole revised manuscript. (line 26, 27, 29, 56, and 131,etc)

4. Grammar and style

1. Improve sentence flow and eliminate awkward phrasing.

Response: Thanks for your reminder. We have revised the sentences in the article to ensure their fluency.

2. Clarify confusing logic. E.g. P>0.05 was significant. This contradicts standard significance thresholds. In other words, this does not signify significance, yet the statement claims the opposite. Confirm the intended meaning and adjust accordingly.

Response: Thank you for your reminder. We found that this is a typo, and we have changed "p>0.05" to "P<0.001" in the revised manuscript. (line 151)

Drug susceptibility

1. Manufacturer name error. “Merrier” should be corrected to bioMérieux.

Response: Thanks for your reminder. We have revised it. (line 156)

2. Redundancy and flow. The phrase ‘Escherichia coli exhibited high sensitivity to. Most antibiotics with rates exceeding 80% and some surpassing 90%...” is a bit repetitive. Group antibiotics into classes for better readability.

Response: Thank you for this constructive suggestion. We have revised the text to: The E. coli isolates exhibited notable susceptibility (>80%) to a wide range of antibiotic classes, such as aminoglycosides, cephalosporins, and carbapenems. Particularly, imipenem (IPM) and ertapenem (ETP) displayed remarkable efficacy. (line 157-159)

3.Scientific tone

1. Change drug sensitivity to antimicrobial susceptibility for precision.

Response: Thank you. We have systematically replaced “drug sensitivity” with “antimicrobial susceptibility” throughout the manuscript to align with standard clinical microbiology terminology (CLSI M100 guidelines). (line 153)

2. State clearly which findings require clinical caution e.g. low fluoroquinolone sensitivity.

Response: Thank you for this constructive suggestion. In the last sentence of the chapter describing each antibiotic sensitivity, we have revised it to remind clinical quinolones that have the lowest sensitivity. (line 165-166)

3. Ensure that drug abbreviations (e.g., AMK, ETP) are defined.

Response: Thank you. We have standardized several antibiotic abbreviations based on the latest CLSI M100 guidelines. For example, cefuroxime sodium (CXM), imipenem (IPM), etc. (line 159, 161, 165, etc)

4. Highlight any multidrug resistance patterns briefly.

Response: Thank you for your professional advice. We have briefly introduced CRE in the revised manuscript. (line 167-171)

DISCUSSION

1. Grammar and language use

Response: Thank you for your careful review and valuable suggestions. We have revised the sentences in the article to ensure their fluency.

1. Rewrite the multiple redundant constructions

Response: Thank you for reminding us that we have appropriately deleted the redundant content in the "discussion" section.

2. Inconsistent verb tenses (e.g. it was found that… to is...)

Response: We have revised the text to ensure all descriptions of our study’s results consistently use past tense (e.g., changing 'is' to 'was' in line 190).

3. Change this phrase … “bacteria detection time to e.g. time to detection” for clarity.

Response: Thank you for your insightful comment. We have revised this phrase throughout the manuscript. (line 193)

2. Scientific tone and structure

Change the informal expressions to professional tone e.g. face more difficult anti-infection….

Response: Thank you. We have changed “face more difficult anti-infection….” to “lead to longer anti-infection treatment”. (line 207)

3. Statistical reporting. Use consistent phrasing for P-values and avoid redundancy.

Response: Thanks for your professional advice. We have rewritten the "Statistical" section to use consistent wording for P-values. (line 107-111)

4. Content clarity

1. The connection between biofilm formation and antibiotic resistance could be more directly and clearly explained.

Response: We have revised the manuscript to provide a more direct and explicit explanation. (line 245-251)

2. Clarify the contrast between your findings and previous literature where appropriate.

Response: Thank you for your professional advice. In the article discussion, we have added our findings and previous comparisons. (line 225-229)

Did you encounter any limitation in this study? If yes, indicate them.

Response: We appreciate the reviewer's opportunity to address limitations. Key constraints include: The sample source was relatively single, and there was no data from other medical institutions in the region. There was no description of how E. coli enters the urinary system from the digestive system. We've put it all in the revised manuscript. (line 263-272)

Reviewer #2:

1. Re-frame the title. Avoid the repeated use of characteristics

Response: Thanks for. We have changed the title as “Clinical epidemiological characteristics and antibiotic sensitivity of Escherichia coli urinary tract infection”.(line1-2)

2. 2nd paragraph, Line4…” 40% of hospital infections” - do you mean hospital acquired infections? Clarify

Response: Thank you for your professional advice. We have changed "hospital infections" to "hospital-acquired infections" in the revised manuscript. (line 56)

3. Same paragraph: The most common gram-negative bacteria is Enterobacteriaceae, mainly E. coli, and Staphylococcus is mainly among Gram-positive bacteria. Replace ‘is’ with are. Uropathogenic Escherichia coli (UPEC) is a group of parenteral pathogenic Escherichia coli should be italicized

Response: Thank you for your kind reminder. We have made the changes as your requested. (line 63-64)

4. The subtitle “Experimental procedure” is perhaps more appropriately expressed as laboratory testing

Response: Thanks for your suggestion. We have changed "Experimental procedure" to "Sample culture and antibiotic sensitivity testing of isolates" in the revised manuscript. (line 79)

5. Still under the same subtitle…“using the VITEK 2 Compact system of the French company Merrier).” Shorten it like......VITEK 2; Merrier). Also apply to other parts of the manuscript where this occurs.

Response: We thank the reviewer for this helpful suggestion to improve conciseness. We have revised all instances of the instrument description according to your recommendation. (line 84-85)

6. Extended-Spectrum β-Lactamases- first use insert (ESBL) after it

Response: Thanks for your suggestion. We have inserted “(ESBL) “where it first appeared in the revised manuscript. (line 87)

5.“Hospitalization time” define; explain

Response: We revised the manuscript to annotate "Hospitalization time" below Table 1 and standardized its use throughout the text with the following annotation: Hospitalization time refers to the total duration (in days) from hospital admission to discharge for a single clinical episode, excluding time in emergency departments or outpatient clinics prior to formal admission. (line 128-130)

6. “in-hospital infection and out-of-hospital infection.” Define these terms.

Response: Thanks for your suggestion. We have changed “in-hospital infection” to “hospital-acquired infection” and “out-of-hospital infection” to “non-hospital acquired infection” in the full manuscript. The first paragraph in the section “Hospital-acquired infection and Non-hospital acquired infection” is defined and interpreted according to references. (line 132-143)

9. “characteristics of the infected population (including gender, age, hospitalization time, etc.)” Time of the year) not given. This will give temporality to the epidemiological data given

Response: Thank you for your reminder. we have revised the manuscript as follows: The clinical data of all cultured E. coli patients (including gender, age, hospitalization time, etc.) were collected and analyzed in the whole year of 2023. (line 91-93)

10. Bacterial culture- could be changed to: Sample culture and antibiotic sensitivity testing of isolates

Response: Thanks for your suggestion. We have revised “Bacterial culture” to “Sample culture and antibiotic susceptibility testing of isolates” throughout the manuscript to better reflect our workflow. (line 79)

11. “Research on bacterial culture and drug sensitivity test methods” – Delete

Response: Thanks for your suggestion. We have removed it.

12. Under subtitle Statistical analysis “This study uses SPSS23.0” uses should be in the past tense. This should be applied across the methods section. English improvement is generally needed.

Response: Thanks for your kind reminder. We have rewritten the statistics section and used the past tense grammatically. (line 107-111)

13. Checkout time in Table 1 should be defined

Response: Thank you for your kind reminder. We have defined “Checkout time” in section “General situation of research subjects”. (line 123-125)

14. Under the subtitle “In-hospital infection and out-of-hospital infection” the first statement is a repetition. Delete

Response: Thanks for your suggestion. We have removed it.

15. Subtitle “A Drug susceptibility of 401 cases of Escherichia coli”: make it simpler. As in: Drug susceptibility of Escherichia coli isolates

Response: Thanks for your suggestion. We have changed “A Drug susceptibility of 401 cases of Escherichia coli” to “Antimicrobial susceptibility of E. coli isolates”. (line 153)

16. May wish to reframe “In contrast, quinolones like levofloxacin (LEV) exhibited the lowest sensitivity rate at 21.9%, warranting heightened attention from clinicians.” As: “Quinolones such as levofloxacin (LEV) exhibited the highest resistance rate to our isolates rate at 78.1% warranting heightened attention from clinicians.” To emphasize the resistance

Response: Thank you. It has been revised in the revised manuscript according to your description. (line 165-166)

17. Table 2: Clinical characteristics of 401 patients under different infection conditions. Delete” under different infection conditions”. Compute and add %s in the table

Response: Thanks for

---

## [Decision Letter · Decision Letter 1]

19 Sep 2025

Dear Dr. Wen,

Thank you for submitting your manuscript to PLOS ONE. After careful consideration, we feel that it has merit but does not fully meet PLOS ONE’s publication criteria as it currently stands. Therefore, we invite you to submit a revised version of the manuscript that addresses the points raised during the review process.

We look forward to receiving your revised manuscript.

Kind regards,

Mabel Kamweli Aworh, DVM, MPH, PhD. FCVSN

Academic Editor

PLOS ONE

Journal Requirements:

Additional Editor Comments: 

In addition to addressing the reviewers' comments, please revise the abstract to follow a structured format with the following sections: Introduction, Methods, Results, and Conclusion. The final paragraph of the abstract’s Introduction section should clearly articulate the study’s aim.

Reviewers' comments:

Reviewer's Responses to Questions

**Comments to the Author**

Reviewer #1: All comments have been addressed

Reviewer #2: All comments have been addressed

2. Is the manuscript technically sound, and do the data support the conclusions?

Reviewer #1: Yes

Reviewer #2: Yes

3. Has the statistical analysis been performed appropriately and rigorously?

Reviewer #1: Yes

Reviewer #2: N/A

4. Have the authors made all data underlying the findings in their manuscript fully available?

Reviewer #1: Yes

Reviewer #2: Yes

5. Is the manuscript presented in an intelligible fashion and written in standard English?

Reviewer #1: Yes

Reviewer #2: Yes

Reviewer #1: The authors have addressed all the comments; corrected and incorporated the suggested comments and the manuscript reads better and is technically sound. Therefore, I recommend this manuscript.

Reviewer #2: The authors have responded to the issues raised.

While the term "non-hospital acquired infection" now being used is OK, changing it to the commonly used community acquired infection will make the paper more searchable and visible as well as increase comparability.

Though they did include limitations as requested, the first limitation stated can be deleted: since the apparent aim of the study is to study "the clinical characteristics of Escherichia coli in urinary tract infection and guide empirical treatment," studying only Escherichia coli derived from the urinary tract infection as opposed to from other organs cannot be a limitation. The remaining limitations can be improved for better readability.

If permitted by the editors, structuring the abstract will give it more clarity and readability especially making clear the aim of the study.

**Do you want your identity to be public for this peer review?** For information about this choice, including consent withdrawal, please see our Privacy Policy

Reviewer #1: **Yes: ** Foluso Ayobami Atiba

Reviewer #2: No

---

## [Author Response · Author response to Decision Letter 2]

7 Oct 2025

Reviewer #1:

The authors have addressed all the comments; corrected and incorporated the suggested comments and the manuscript reads better and is technically sound. Therefore, I recommend this manuscript.

Response:Thank you. Although no additional comments were raised by you during the present revisions, in order to make the readers more readable, we have made changes to the details of the article and improved the manuscript as much as we can. We appreciate your comment on our paper. The revised manuscript has been resubmitted. And hope to consider publishing in PLOS ONE.

Reviewer #2

While the term "non-hospital acquired infection" now being used is OK, changing it to the commonly used community acquired infection will make the paper more searchable and visible as well as increase comparability.

Response:Thank you for your suggestion, we have replaced all the "non-hospital acquired infection" in the full revised manuscript with "community acquired infection".(line27,29,94,134,etc,table2,4)

Though they did include limitations as requested, the first limitation stated can be deleted: since the apparent aim of the study is to study "the clinical characteristics of Escherichia coli in urinary tract infection and guide empirical treatment," studying only Escherichia coli derived from the urinary tract infection as opposed to from other organs cannot be a limitation. The remaining limitations can be improved for better readability.

Response:Thank you for your professional advice, we have removed the first limitation, and the explanation you gave has benefited us a lot. To make readers more readable, we have improved the remaining limitations.(line267-273).

If permitted by the editors, structuring the abstract will give it more clarity and readability especially making clear the aim of the study.

Response:Thank you for your proposal. We have modified the abstract section according to the structure of “Introduction, Methods, Results, and Conclusion” according to the editor’s requirements and clarified the purpose of this study at the end of “Introduction”.(line16-18)

Journal Requirements:

Response:Thank you for your enthusiastic reminder. We carefully checked the reference list and found that there were indeed inappropriate articles being cited, and we have replaced them based on the content of the article. (References from Articles 7, 11, 27)�line301-304,316-319,375-378

---

## [Editor Report · Decision Letter 2]

14 Oct 2025

Dear Dr. Wen, 

Thank you for submitting your manuscript to PLOS ONE. After careful consideration, we feel that it has merit but does not fully meet PLOS ONE’s publication criteria as it currently stands. Therefore, we invite you to submit a revised version of the manuscript that addresses the points raised during the review process.

We look forward to receiving your revised manuscript.

Kind regards,

Mabel Kamweli Aworh, DVM, MPH, PhD. FCVSN

Academic Editor

PLOS ONE

Journal Requirements:

**Additional Editor Comments:**

Thank you for your thoughtful revisions to the manuscript. Please review the abstract to ensure clarity and smooth flow. Kindly remove the subtitle “Limitations” and incorporate that content into the final paragraph of the Discussion section. Also, check the spelling of *E. coli* —make sure there is a space between “E.” and “coli.” Lastly, please add a period "." to the final sentence of the Discussion section.

Thank you again for your efforts.

---

## [Author Response · Author response to Decision Letter 3]

21 Oct 2025

Thank you for your reminder. We have made slight modifications to the abstract to make the content clearer and the sentences smoother (line17,18, 26-27,31,38-43). Secondly, we have also deleted the subtitle "Limitations" and included it in the last paragraph of the Discussion section (line255-261). We also checked the spelling of all "E. coli" in the entire manuscript, and four spellings were found to be inconsistent with the specification (line37, 230,255,258), and a space was added between "E" and "coli"; Finally, a period ". " was added to the last sentence of the original Discussion part(line254). At last, on behalf of our research team, I would like to express my gratitude for your hard work and wish you success in your work and a happy mood.

---

## [Editor Report · Decision Letter 3]

29 Oct 2025

Clinical epidemiological characteristics and antibiotic sensitivity of Escherichia coli urinary tract infection

PONE-D-25-28510R3

Dear Dr. Wen,

We’re pleased to inform you that your manuscript has been judged scientifically suitable for publication and will be formally accepted for publication once it meets all outstanding technical requirements.

Kind regards,

Mabel Kamweli Aworh, DVM, MPH, PhD. FCVSN

Academic Editor

PLOS ONE
---

## [Editor Report · Acceptance letter]

PONE-D-25-28510R3

PLOS ONE

Dear Dr. Wen,

I'm pleased to inform you that your manuscript has been deemed suitable for publication in PLOS ONE. Congratulations! Your manuscript is now being handed over to our production team.

Kind regards,

on behalf of

Dr. Mabel Kamweli Aworh

Academic Editor

PLOS ONE